# Exploring Patient Adherence to Post-Surgical Follow-Up in Pelvic Endometriosis

**DOI:** 10.3390/medicina59071210

**Published:** 2023-06-28

**Authors:** Tamas Szabo, Melinda-Ildiko Mitranovici, Andrada Crisan, Ioana Marta Melinte, Teodora Cotrus, Vlad Tudorache, Liviu Moraru, Raluca Moraru, Romeo Micu

**Affiliations:** 1Doctoral School of Medicine and Pharmacy, George Emil Palade University of Medicine, Pharmacy, Science, and Technology of Târgu Mureș, 540142 Targu Mures, Romania; szabo_tamas91@yahoo.com (T.S.); crisanandrada@yahoo.com (A.C.); ioanammelinte@gmail.com (I.M.M.); cotrus.teodora12@gmail.com (T.C.); 2Department of 1st Gynecology Clinic, Emergency County Hospital Targu Mures, 50 Gheorghe Marinescu Street, 540136 Targu Mures, Romania; 3Department of Obstetrics and Gynecology, Emergency County Hospital Hunedoara, 14 Victoriei Street, 331057 Hunedoara, Romania; 4Department of 2nd Gynecology Clinic, County Clinical Hospital Targu Mures, 6 Bernády György Square, 540072 Targu Mures, Romania; vladtudorache1994@gmail.com; 5Department of Anatomy, “George Emil Palade” University of Medicine, Pharmacy, Sciences and Technology, 540142 Targu Mures, Romania; liviu.moraru@umfst.ro (L.M.); raluca.moraru@umfst.ro (R.M.); 6Department of Human Assisted Reproduction of 1st Gynecology Clinic, University of Medicine and Pharmacy “Iuliu Hatieganu”, 400347 Cluj-Napoca, Romania; romeomicu@hotmail.com

**Keywords:** endometriosis, laparoscopy, diagnosis, management, follow-up adherence, malignancy

## Abstract

Endometriosis is a significant cause of disability that affects 5–10% of reproductive-aged women. Laparoscopy with histological confirmation is the gold standard in establishing the diagnosis as therapeutic management surgery is addressed to a certain category of patients. The objective of this study was to assess patient adherence to follow-up after surgery for endometriosis as the primary endpoint, pain symptoms, quality of life, mental health, and fertility as the secondary endpoints. We have analyzed patients’ adherence to follow-up after surgery for endometriosis after 1, 3, 5, and 7 years. Out of the 2538 total number of surgeries, 453 patients replied just to the first questionnaire (group A), 528 to the first and second (group B), and only 356 carried out the entire follow-up schedule. General health was significantly lower for group A (46.6 vs. 56.4) but with no statistical difference in the post-surgical improvement in both groups. Pain level score improvement was lower for group A (10.5 vs. 18.8), which is statistically significant. In this light, laparoscopy still remains the gold standard in diagnosis only. Furthermore, no malignancy was discovered. The mental component was improved by laparoscopy based on SF-36 in group B. Studies on patient preference for surgery versus alternative treatment are needed.

## 1. Introduction

Endometriosis is a gynecological disorder characterized by the growth of endometrial tissue outside of the uterus, which affects 5–10% of reproductive-aged women [1]. Endometriosis is a significant cause of disability. According to studies in specialized literature, endometriosis has a negative impact on quality of life [1]. It is a progressive disease, and the patterns of symptoms may change over time due to changes in the type of endometriosis lesions [2].

There are three types of endometriosis described in the literature: ovarian, superficial peritoneal, and deep infiltrating endometriosis [3,4].

The pathogenesis of endometriosis is not fully understood, but it is considered to be a hormonally responsive tissue similar to the endometrium. Nevertheless, there is evidence in the literature of symptomatic endometriosis occurring in menopause, where genetic involvement is more likely [5]. It is very rare in patients who are not taking hormone replacement therapy, and it should be less active in menopause as compared to the reproductive years [5].

The most widely accepted pathogenic mechanism of endometriosis is menstrual reflux. However, due to the heterogeneity of the disease manifestations, multiple theories and hypotheses have been proposed, including dislocation of the basal endometrium, metaplasia of peritoneal mesothelial cells or endometrial stem cells, lymphatic and hematogenic spread, and exposure to certain agents [5,6]. Phenotypically, endometriotic cells differ from normal endometrial cells due to environmental, immunological, inflammatory, or other predisposing factors [5,7]. The genetic and epigenetic theories emphasize that endometriotic cells undergo genetic and epigenetic transformations, with changes in gene function not necessarily involving changes in the DNA sequence [5]. Environmental toxins are considered potential molecular causes of progesterone resistance [1]. Epigenetic modifications affect the PCR gene transcription, and oxidative stress is involved. Deep endometriosis is more resistant [3,8]. The most commonly involved sites are the ovaries, uterosacral ligaments, cul-de-sac, fallopian tube, sigmoid, colon, appendix, rectovaginal septum, cervix, vagina, bowel, urinary bladder, ureter, and umbilicus. Rare locations include bones, the central nervous system, lungs, and eyes [6].

Most commonly, the symptoms consist of pain or infertility, which is also the reason for diagnostic laparoscopic intervention [2]. The prevalence of endometriosis is 30–80% in women with pelvic pain and 20–50% in women with infertility [6]. One-third of women are asymptomatic [6]. The pain can manifest as dysmenorrhea, dyspareunia, and dyschesia but also as non-cyclic chronic pain, sometimes associated with an ovarian mass. It seems that in cases of adolescent-onset disease, cyclic menstrual pain is present in only 9.4% of cases [2,9]. Patients are driven to seek medical care almost solely through life-impacting pain symptoms [2]. Nociceptive neuropathia is influenced by stress, inflammation, hormones, and the peripheral and central nervous system. Endometriotic lesions themselves send nociceptive signals to dorsal root spinal cord neurons. In addition, anxiety, depression associated with debilitating pain, the amplitude of pain perception, and chronic fatigue—whose mechanism is not completely understood—make endometriosis a public health problem and a major economic burden [1,9].

Despite the significant impact of endometriosis symptoms, there are often delays in the diagnosis and treatment of the condition, with women diagnosed during adulthood experiencing a three times longer delay compared to those diagnosed in adolescence in receiving a correct diagnosis. This delay may be due to fear of stigma associated with gynecological complaints or a lack of education [2].

The management of endometriosis can be approached either surgically or medically. Estrogen exposure is a major risk factor for this condition, and current treatments aim to suppress its production [1]. Nonsteroidal anti-inflammatory drugs and/or combined contraceptive pills are the first line of therapy. However, it has been observed that one-third of patients who take hormone medication for pain do not experience any symptom improvement, likely due to the decreased responsiveness of endometriosis-associated pain to hormonal therapies [2]. Furthermore, the use of agonists or antagonists is not cost-effective and may not necessarily be efficient [10]. As a result, attention has shifted towards the use of selective progesterone and estrogen modulators, antiangiogenic factors, immunomodulators, and aromatase inhibitors [6]. This has led to the need for alternative therapies, and operative laparoscopic intervention may be considered superior to diagnostic laparoscopy in alleviating pain, but the associated risks and complications must be taken into account [9,11,12].

For the same reason, alternative diagnostic methods must be sought, even if laparoscopy is the golden standard, along with histopathology. Ultrasound, CT scan (Computer Tomography), and MRI scan (Magnetic Resonance Image) do not bring images specific to endometriosis. Reliable biomarkers are being sought so far, with CA-125 being the most studied, but it is not specific to the pathology. Extended additional studies are needed in this regard [13,14,15].

## 2. Materials and Methods

The objective of this study was to assess patient adherence to follow-up after surgery for endometriosis as the primary endpoint, with changes in pain symptoms, disease recurrence, quality of life, mental health, and fertility as secondary endpoints. Through this questionnaire, it can be determined to what extent the surgical intervention is useful as therapeutic management or if it remains only the golden standard in the diagnosis of endometriosis, being able to be used in certain cases as therapeutical management. Furthermore, through follow-up, we can detect possible malignancies.

### 2.1. Study Population

Women included in this retrospective analysis of prospectively collected data were managed from June 2009 to June 2019 for pelvic endometriosis.

The management of these individuals took place within one of the 15 establishments engaged with the CIRENDO database—a progressive cohort under the North–West Inter Regional Female Cohort for Patients with Endometriosis; NCT02294825—funded by the G4 Group (which encompasses the University Hospitals of Rouen, Lille, Amiens, and Caen in France, and the Association of Endometriosis Surgeons known as ROUENDOMETRIOSE in France).

The CIRENDO database adopted specific inclusion criteria: only women whose diagnosis of endometriosis was substantiated through surgical investigation and histological analysis were incorporated into the database.

There were also exclusion criteria to consider any missing details in the electronic database, or an incomplete questionnaire would result in the exclusion of the subject from the study.

Data was gathered from both surgical and histological documentation, in addition to self-administered questionnaires filled out prior to surgery. These questionnaires encompassed various aspects such as medical history, sociodemographic information, fertility status, recurring and persistent pain, issues with sexual activities and discomfort during intercourse, gastrointestinal problems, and overall quality of life.

Intraoperative findings and immediate postoperative management were recorded using questionnaires filled in by the surgeon. Follow-up self-questionnaires were addressed to patients at 1, 3, 5, and 7 years after the surgery, including questions about endometriosis-related symptoms and fertility outcomes. The roles of data logging, maintaining patient communication, and overseeing patient follow-up were managed by a professional clinical research technician. The forward-looking data collection and interpretation received authorization from French regulatory bodies, namely the CNIL (Commission Nationale de l’Informatique et des Libertés) and the CCTIRS (Comité Consultatif pour le Traitement de l’Information en matière de Recherche dans le domaine de la Santé), prior to being incorporated into the CIRENDO database. Patients were preoperatively informed about the goal and the risks of surgical management of endometriosis. All patients gave written informed consent to participate in the study.

The study was approved by the Univ”rsit’ of Medicine, Pharmacy, Science and Technology George E. Palade Targu Mures Institutional Ethics Committee for Non-Interventional Research (no. 2336/17 May 2023).

### 2.2. Assessment of Symptoms during Follow-Up

The intensity of pain symptoms was evaluated before surgery and during the follow-up. Questionnaires were used to evaluate chronic pain, dysmenorrhea, dyspareunia, quality of life (Short Form 36, SF 36), mental health, emotional well-being, self-image, and urinary symptoms.

A validated language version of Kess (Knowles-Eccersley-Scott-Symptom Questionnaire) and a GastroIntestinal Quality of Life Index CIQLI questionnaire were used for intestinal symptoms and change in bowel habit.

Upon completing their surgery, the women were asked to evaluate their overall satisfaction with the provided treatment and determine whether there was a recurrence of pain or the disease by responding to a series of questions.

### 2.3. Statistical Analyses

Statistical analysis was performed using the Stata 16.0 software (StataCorp LP, College Station, TX, USA). Univariate analysis compared patient characteristics, clinical history, baseline complaints, intraoperative data, and postoperative outcomes. Fischer’s exact test was used to compare qualitative variables and analysis of variance and *x*^2^ test was used to compare continuous variables. The regression analysis was performed using the Cox regression test. A *p* value of less than 0.05 was considered statistically significant.

## 3. Results

We have analyzed patients’ adherence to follow-up after surgery for endometriosis after 1, 3, 5, and 7 years. Out of 2538 total surgeries, 453 patients replied just to the first questionnaire, 528 to the first and second, 445 to the first three questionnaires, and 356 carried out the entire follow-up schedule. So, group A includes 445 patients, those who answered the questionnaire before surgery and one year after surgery and then lost the follow-up. Group B includes 528 patients who answered at least two questionnaires after surgery.

The mean age of our patient was 32.52, with an SD Standard deviation of 6.87. We have assessed their social background, level of education, medical history, and wish for future pregnancy.

Most patients undergoing surgery for endometriosis were employed and in a romantic relationship, with a tendency towards marriage for the group that adhered to the long-term follow-up.

Dysmenorrhea and general pelvic pain are common symptoms of endometriosis regardless of age at diagnosis. They took hormonal medication for pain but did not have any improvement. General pelvic pain made it difficult to participate in work or normal daily activities and social life before surgery. Exercise was also impacted more severely than expected. Urinary and gastrointestinal symptoms were also present [Table 1].

### CIRENDO

All patients answered the same questionnaire before and after the operation and during the follow-up.

There were no significant differences in regard to mental status and psychiatric disorders between the evaluated groups.

Most patients that answered postoperative questionnaires were diagnosed with stage IV rAFS endometriosis [Table 1].

There was no significant difference in follow-up for patients suffering from primary or secondary infertility.

An overall 51.01% of patients desired a pregnancy prior to surgery, with no difference between the groups evaluated.

There were no significant differences in obtaining spontaneous or assisted pregnancies and deliveries between the groups at the 1-year postoperative evaluation [Table 1].

Out of those who desired to conceive, 1/5 patients at the 1-year postoperative evaluation, respectively 1/4 patients at the 3-year postoperative evaluation have obtained a pregnancy, which makes the surgical intervention useful to some extent in obtaining pregnancy. However, it is difficult to evaluate without knowing other procedures of assisted human reproduction techniques that could be used by patients during the follow-up period [Table 1].

In the same way, those who were not lost to follow-up showed similar improvement in postoperative symptoms, including in general health and physical condition, but these were without statistical significance, and in the answers to the KESS questionnaire, they reported dissatisfaction [Table 2]. There were no cases of malignancy in the groups.

Here is the comparison of the two groups to understand what caused the loss of adherence to the follow-up and also to understand the way the surgery brought improvements rather than the diagnosis [Table 3].

We compared patients that were lost to follow-up after one year post-surgery (group A) to patients who adhered throughout the postoperative monitoring (group B) [Table 3].

During menstrual cycles, pain levels, the need for antalgic treatment, and the association with urinary symptoms were reported to be higher and more frequent before surgery in group A (urinary symptoms 34% vs. 23.6%, *p* < 0.05; pain intensity with no treatment 8.48 vs. 7.89, *p* < 0.05; pain intensity with treatment 4.84 vs. 4.16, *p* < 0.05). Moreover, the decrease in intensity of pain reported after surgery between the two groups was significantly higher for patients in group A (overall pain 63.8% from 95.6% to 31.8% vs. 56% from 93.5% to 37.5%, *p* < 0.05; the need for pain treatment 21.7% from 88.6 to 66.9% vs. 12.5% from 87.3 to 74.8%, *p* < 0.05; pain intensity with no treatment 2.11 meaning 8.48 minus 6.37 vs. 1.9 meaning a drop from 7.89 to 5.99, but not significantly *p* > 0.05; pain intensity with treatment 1.71 again 4.84 minus 3.13 vs. 1.28 mean intensity dropped from 4.16 to 2.88, but not statistically significant *p* > 0.05), on the pain scale 0 meaning no pain and 10 meaning extreme pain. However, patients in group B reported a decrease in pain levels during menstrual cycles during the follow-up (after 1 year, after 3 years, after 5 years, and after 7 years), follow-up being important for this reason [Table 3].

Pain during intercourse was reported significantly more often in group A compared to group B (prior to surgery. 78.3% vs. 68.5%; 1-year post surgery 39.5% vs. 37.0%) [Table 2 and Table 3], but not statistically significant, but the limitations during intercourse were statistically significant because of the pain *p* < 0.05. The intensity was also significantly higher for group A with levels averaging higher on the pain scale with and without treatment (no treatment: prior to surgery 6.38 vs. 5.76, 1 year post surgery 5.28 vs. 4.64; with treatment: prior to surgery 4.61 vs. 4.13 vs. 3.88 vs. 3.74 but again not statistically significant).

Over half of the patients who adhered to follow-up every 1 year for 3 years after surgery had digestive endometriosis involvement (58.94%, respectively 52.65%). In comparison, 35.67% of patients that were not lost to follow-up had digestive involvement after 7 years of follow-up. Moreover, patients in group A had a higher KESS questionnaire score prior to surgery (12.7 vs. 11.2 with a cut-off at 11 that indicates constipation) with statistical significance and a significantly lower GIQLI score (prior to surgery 83.9 vs. 93.1, 1 year after surgery 98.9 vs. 105.2) compared to group B.

However, there are no differences for patients with urologic involvement none of the patients from groups A or B developed malignant transformation.

SF36 score was used to assess the health status of the patients and to measure the impact the surgery had on their quality of life:The physical functioning segment had a significantly lower score for group A compared to group B before surgery (60.8 vs. 71.7) with lower postoperative improvement (8 vs. 7.6)Role limitations due to physical functioning were reported at 27.6 for group A and 46.4 for group B prior to surgery, with higher postoperative improvement in the case of group A (16.9 vs. 13.7)(47.3 vs. 53.5; 46.6 vs. 56.4), but with no statistical difference in post-surgical improvement.Pain level score improvement was lower for group A, 10.5 (60.0–49.5 number of patients) vs. 18.8 (67.1–48.3 number of patients), and was statistically significant.There were no differences in the score increase for energy levels between the two groups; however, there were significantly lower scores both before and after surgery for group A.Social functioning was reported to be significantly lower for group A (before surgery, 49.2 vs. 54.5; 1 year after surgery, 58.2 vs. 67.1).Role limitations due to emotional involvement before and after surgery had overall lower scores for group A (before surgery, 44.2 vs. 52.4; 1 year after surgery, 57.6 vs. 64.9) but with no statistical difference in the post-surgical improvement.

## 4. Discussion

Even if the decrease in intensity of pain reported after surgery between the two groups was significantly higher for patients in group A, as the percentage and number of patients and pain level score improvement were lower for group A (mean intensity without treatment 6.37 in group A vs. 5.99 in group B, mean intensity with treatment 3.13 versus 2.88) and statistically significant, during menstrual cycles or chronic pain. In addition, the physical functioning segment had a significantly lower score for group A compared to group B’s role limitations due to it. General health and social functioning were reported to be significantly lower for group A. Role limitations due to emotional involvement after surgery had overall lower scores for group A. There are no significant differences in urologic symptoms between groups A and B, and no malignancy was discovered.

Patient education and medical professional training are necessary to fully understand the nature of this pathology [2]. Our research has numerous strengths, including a large cohort of patients, surgical diagnosis of endometriosis in a tertiary center, standardized and validated data, detailed information on clinically evaluated symptoms, and a variety of pain types, locations, and patterns. Our study was designed to emphasize the importance of timely surgery for this condition, surgery being the gold standard, along with histopathological findings in diagnosis.

Study Limitations: Our study had several limitations, including under sampling of women with infertility, unrevealed rASRM (the revised American Society for Reproductive Medicine) scores, and potential selection bias. Additionally, we did not consider ethnic criteria, economic background, urban/rural residence, or BMI. Furthermore, no studies have provided definitive evidence on when surgery should or should not be performed due to the lack of conclusive data. Furthermore, regarding patient adherence to the follow-up, we can only speculate that patient dissatisfaction is the main reason for the loss of long-term follow-up.

In cases of repeat surgery, recurrent pain was the primary indication for re-operation. Our study found that surgery did not significantly improve overall pain levels, especially in group A, where they exceeded group B in the number and percentage of patients 1 year after surgery, but not as pain level scale. There is a gap in long-term pain outcomes.

This is confirmed by the lack of follow-up adherence of the patients in group A, probably due to an improvement that does not correspond to their expectations.

In addition, the live birth rate was improved in our study, which is confirmed by Matheo Leonardi’s research [16], but there is a bias because we do not know the methods of assisted human reproduction used during this period.

Mental components improved by laparoscopy based on SF-36 and were also better in group B for those who were monitored for a longer period of time, which confirms the data from the literature [16,17,18,19].

Studies on patient preference for surgery versus alternative treatment are needed. In addition, there is a poor correlation between the severity of the symptoms and disease grade [16].

The heterogeneity of the pathology imposes the complexity of therapeutic management [17,18], and our study confirms the need for alternative therapy to surgery.

It is worth noting that malignant transformation is extremely rare in endometriosis, and no cases were described in this study, despite the similar pathogenesis and behavior of endometriosis and cancer. This genetic and epigenetic instability is consistent with the infrequent occurrence of malignant transformation of endometriosis [5,20,21,22,23,24].

The clinical guidelines from ESHRE, rASRM, and ACOG (American College of Obstetrics and Gynecology) recommend hormonal treatment as the first-line therapy [3,17]. However, the absence of controlled studies comparing the effectiveness of progesterone to contraceptive pills leaves room for interpretation. Hence, there are doubts regarding the usefulness of progestins in controlling pelvic pain [17,25,26]. The heterogeneity of endometriosis manifestations explains the variable response to medication and resistance to hormonal treatment [5,27,28]. The underlying cause is likely dysregulation of the progesterone signaling pathway and gene networking in the endometriosis tissue [1]. Additionally, deep endometriosis is resistant to progestin treatment [3,12,29]. In the case of deep endometriosis, fibroblast and myofibroblast trans-differentiation contribute to collagen production and fibrogenesis with entrapment of nerve fibers, which explains chronic pain symptoms [9,30].

The ESHRE guidelines recommend surgery for drug-resistant pain [3,9], and robotic surgery is an option [31,32,33,34]. Recurrent pain without evidence of recurrent disease can occur, and this is explained by the impact on the central nervous system and altered perception, with an increase in neuronal responsivity and emotional pain pathway and a large individual variation [18,29]. Therefore, surgery has limitations. Additionally, surgical methods can reduce ovarian reserve, and in cases of infertility, assisted reproductive techniques are preferred. In such cases, exercise and physical activity, including cardiovascular fitness or yoga, have been suggested. The outcomes need to be measured using reliable and validated tools [19,32].

Various treatments are currently used, including GnRH analogs or antagonists, anti-inflammatory agents, antiangiogenic agents, agents that induce apoptosis, agents that reduce oxidative stress, and third-generation nonsteroidal anti-inflammatory agents such as letrozole, monoclonal antibodies, and aromatase inhibitors [1,9,18,35]. However, these treatments may have many side effects, such as hot flashes, decreased bone density, depression, and impairment of mental health. Therefore, it is important to provide appropriate counseling to patients. A comprehensive overview of the efficacy and side effects of all available therapies is needed [12].

A novel therapeutic approach for endometriosis should be based on the disease’s pathogenesis [1]. Several target molecules and genes are currently being studied, including those that modulate PGR target genes, such as FOXO1A and hox-A1 [1]. Osteopontin is a molecule that is decreased in endometriosis and implicated in infertility, and Progesterone Receptor Gene polymorphism is being studied as a potential target [1,20,23]. Researchers are also investigating antiangiogenic therapies, including TNP-470, a fumagillin analog; endostatin, a proteolytic fragment of collagen with endogenous antiangiogenic activity; anginex, a synthetic peptide that stops the growth of blood vessels; and statins with antioxidant effects, and antibodies against TNF alpha [6,9,21,24]. Icon is a promising therapy for endometriosis, as it destroys endometriotic tissue through vascular disruption. This novel antibody-like immunoconjugate molecule appears to have no apparent toxicity and does not reduce fertility or cause teratogenic effects [28]. Dopaminergic agonists have been shown to reduce angiogenesis by inducing endocytosis of the VEGF receptor 2, while cannabinoids have been found to control pain symptoms [24,28]. As our understanding of the pathogenic mechanisms underlying endometriosis improves, newer targets for treatment could be discovered [6,28].

Hypermethylation of DNA contributes to progesterone resistance, and demethylation agents may be a future research approach [1,13,14]. Furthermore, metformin could regulate the expression of dysregulated miRNA in endometriotic mesenchymal stem cells [1,15,20]. Therefore, innovative drug design, better phenotyping, personalized treatment, and drugs acting beyond PR (progesterone receptor) are unaffected by progesterone resistance are the future. The need for a new strategy and a deeper understanding of this pathology should be the innovation in the management of this disease [8,27].

The scavenging system for iron overload, which is responsible for oxidative stress and inflammation, is being studied as a potential future approach. Retrograde menstruation and bleeding of endometriotic tissue are responsible for the release of hemoglobin into the peritoneal cavity, which has toxic effects with pro-oxidant and pro-inflammatory effects. Continuous delivery of iron to macrophages may overwhelm the capacity of ferritin to store iron. Efforts to scavenge and eliminate Iron could be a potential solution to blocking the progression of endometriosis [12,13,15,21].

All these efforts are justified in reducing the need for surgical re-interventions [30]. Endometriosis surgery is associated with complications and recurrence, and it does not address the underlying pathogenic mechanism of the disease. Additionally, it may have a negative effect on ovarian reserve [30].

Some of the studies showed a slight reduction in health-related quality of life and sexual quality of life after a long-term follow-up [36,37]. Most studies did not report relevant outcomes related to the recurrence of the disease or assessment of pain and quality of life. We highlight the need for long-term follow-up after surgery and standardized reporting of patients’ satisfaction [38]. In the end, a long-term follow-up and medical treatment should be considered [39].

Costs should not be the primary concern in management decisions, although they cannot be disregarded. Surgery can remove endometriotic lesions, but it cannot provide a cure for the disease, and safety concerns remain a priority for future interventions [12,34].

Another concern for the future is finding effective diagnostic methods for endometriosis, as diagnostic laparoscopy with histopathological examination is currently the golden standard, with associated surgical complications. Currently, there is no efficient method to replace surgery apart from imaging techniques such as ultrasound, MRI, CT, and non-specific biomarkers like CA125 [6,13,14,15]. However, the efficacy of these methods is still limited, and pelvic pain recurrence is common [12].

## 5. Conclusions

Individualized treatment is crucial in the management of endometriosis, and surgery remains an important therapeutic option in the absence of a more effective method of diagnosis and treatment. To avoid repeat surgeries and optimize cost-effectiveness, treatment decisions should be tailored to the individual patient’s needs. Even if the decrease in intensity of pain reported after surgery between the two groups was significantly higher for patients in group A, a percentage of patients’ pain level score improvement was lower for group A, which was statistically significant (10.5 vs. 18.8), according to the SF36 score. In this light, laparoscopy still remains the golden standard in diagnosis only. There is a gap in long-term pain outcomes. Because of the lack of adherence to follow-up, we can only speculate that patient dissatisfaction was the cause.

The mental component was improved by laparoscopy based on SF-36 but in the group of those who were monitored for a longer period of time. No malignancy was reported in both groups. Studies on patient preference for surgery versus alternative treatment are needed.

However, making strong recommendations in the absence of high-quality randomized trials can be challenging. Therefore, further research is necessary to develop more effective and personalized treatment strategies for endometriosis.

## Figures and Tables

**Table 1 medicina-59-01210-t001:** Table with patients’ adherence to follow-up after surgery for endometriosis, their marital status, socio-economic status, infertility with pregnancy status during follow-up, and symptoms (Group B).

		S1		S3		S5		S7		
		n	%	n	%	n	%	n	%	*p*
No. patients	Surgeries 2538	453	25.42	528	29.63	445	24.97	356	19.98	
Marital status	Married	152	33.85	178	33.97	163	36.88	149	41.97	<0.000
	Celibate	70	15.59	72	13.74	80	18.10	65	18.31	
	Divorced	11	2.45	12	2.29	16	3.62	14	3.94	
	Widowed	24	5.35	12	2.29	6	1.36	4	1.13	
	Concubinage	180	40.09	232	44.27	166	37.56	118	33.24	
	Other	12	2.67	18	3.44	11	2.49	5	1.41	
Employment	Employed	370	82.96	432	82.29	364	82.54	295	84.05	0.001
	Housewife	16	3.59	14	2.67	20	4.54	15	4.27	
	Student	9	2.02	18	3.43	31	7.03	15	4.27	
	Unemployed	51	11.43	61	11.62	26	5.9	26	7.41	
Mental status	Psych. Dis.	155	27.00	169	29.44	136	23.69	114	19.86	>0.05
	Anxiety	118	27.31	128	29.63	99	22.92	87	20.14	
	Depression	55	24.77	74	33.33	55	24.77	38	17.12	
	Insomnia	79	25.99	89	29.28	79	25.99	57	18.75	
	Panic attacks	35	29.66	34	28.81	33	27.97	16	13.56	
rAFS	I	49	11.19	66	12.74	75	16.93	49	13.96	0.001
	II	82	18.72	84	16.22	86	19.41	49	13.96	
	III	67	15.30	91	17.57	94	21.22	87	24.79	
	IV	240	54.79	277	53.47	188	42.44	166	47.29	
Infertility	Primary	98	62.42	106	55.79	71	52.99	49	47.12	0.09
	Secondary	59	37.58	84	44.21	63	47.01	55	52.88	
Pregnancy status	Desire before surgery	243	53.64	299	56.63	213	47.87	154	43.06	0.064
	PMA after 1 year	42	9.28	44	8.33	38	8.54	24	6.73	>0.05
	Pregnancy after 1 year	92	20.31	89	16.86	96	21.57	76	21.35	<0.05
	Delivery after 1 year	34	7.51	26	4.92	23	5.17	14	3.93	>0.05
	PMA after 3 years			91	17.24	63	14.16	43	12.08	<0.05
	Pregnancy after 3 years			168	31.82	139	31.24	104	29.21	<0.05
	Delivery after 3 years			143	27.08	127	28.54	93	26.12	<0.05
	PMA after 5 years					10	2.23	5	1.4	>0.05
	Pregnancy after 5 years					25	5.62	28	7.87	>0.05
	Delivery after 5 years					21	4.72	25	7.02	>0.05
	PMA after 7 years							1	0.28	>0.05
	Pregnancy after 7 years							14	3.93	>0.05
	Delivery after 7 years							10	2.81	>0.05
Digestive involvement		267	58.94	278	52.65	177	39.78	127	35.67	<0.05
Urologic involvement		76	16.78	78	14.77	47	10.56	54	15.17	>0.05

n—number of patients; %—percentage of patients; S1, S3, S5, S7—the same questionnaire after 1, 3, 5, respectively 7 years of follow-up; PMA—medically assisted procreation.

**Table 2 medicina-59-01210-t002:** Table with patients who lost follow-up after 1 year, comparison before and after surgery (Group A).

Patients Lost to Follow-Up after 1 Year		Before Surgery		S1		S3		S5		S7		
		n	%	n	%	n	%	n	%	n	%	*p*
Menstrual cycles	Pain	432	95.6	135	31.8							<0.05
	Urinary symptoms	145	34.0	26	20.5							<0.05
	Pain treatment	379	88.6	87	66.9							<0.05
	Pain intensity (no treatment)	8.48		6.37								<0.05
	Pain intensity (with treatment)	4.84		3.13								<0.05
Intercourse	Pain	354	78.3	160	39.5							<0.05
	Limitations	334	95.2	134	84.3							<0.05
	Pain intensity (no treatment)	6.38		5.29								<0.05
	Pain intensity (with treatment)	4.61		3.88								>0.05
Pain associated to endometriosis		373	82.9	227	55.1							<0.05
	Pain treatment	259	70.8	117	54.4							<0.05
	Pain intensity (no treatment)	6.68		5.28								<0.05
	Pain intensity (with treatment)	3.79		2.76								>0.05
SF36	Physical functioning	60.8		64.6								>0.05
	Role limitations due to phys. Funct	27.6		44.5								<0.05
	Pain	49.5		60.0								<0.05
	General health	47.3		46.6								>0.05
	Energy/fatigue	32.9		37.7								<0.05
	Social functioning	49.2		58.2								<0.05
	Role limitations due to emo.	44.2		57.6								<0.05
KESS		12.7		10.6								>0.05
Giqli		83.9		98.9								<0.05

n—number of patients; %—percentage of patients; S1, S3, S5, S7—the same questionnaire after 1, 3, 5, respectively 7 years of follow-up (they lost follow-up after 1 year).

**Table 3 medicina-59-01210-t003:** Table for comparing patients who were lost to follow-up after 1 year with patients who were not lost to follow-up.

		Patients Lost to Follow-Up after 1 Year (Group A)				Patients Not Lost to Follow Up (Group B)				
		Before Surgery		S1		Before Surgery		S1		
		n	%	n	%	n	%	n	%	*p*
Menstrual cycles	Pain	432	95.6	135	31.8	330	93.5	132	37.5	<0.05
	Urinary symptoms	145	34.0	26	20.5	76	23.6	15	11.9	<0.05
	Pain treatment	379	88.6	87	66.9	281	87.3	98	74.8	<0.05
	Pain intensity (no treatment)	8.48		6.37		7.89		5.99		>0.05
	Pain intensity (with treatment)	4.84		3.13		4.16		2.88		>0.05
Intercourse	Pain	354	78.3	160	39.5	244	68.5	129	37.0	>0.05
	Limitations	334	95.2	134	84.3	216	92.4	111	87.4	<0.05
	Pain intensity (no treatment)	6.38		5.29		5.76		4.64		>0.05
	Pain intensity (with treatment)	4.61		3.88		4.13		3.74		>0.05
Pain associated to endometriosis		373	82.9	227	55.1	239	67.9	183	52.6	<0.05
	Pain treatment	259	70.8	117	54.4	145	64.4	115	66.9	<0.05
	Pain intensity (no treatment)	6.68		5.28		6.18		5.16		>0.05
	Pain intensity (with treatment)	3.79		2.76		3.66		3.04		<0.05
SF36	Physical functioning	60.8		64.6		71.7		79.3		<0.05
	Role limitations due to phys. Funct.	27.6		44.5		46.4		60.1		>0.05
	Pain	49.5		60.0		48.3		67.1		<0.05
	General health	47.3		46.6		53.5		56.4		>0.05
	Energy/fatigue	32.9		37.7		36.6		43.7		>0.05
	Social functioning	49.2		58.2		54.5		67.1		<0.05
	Role limitations due to emo.	44.2		57.6		52.4		64.9		>0.05
KESS		12.7		10.6		11.2		10.2		<0.05
Giqli		83.9		98.9		93.1		105.2		>0.05

## Data Availability

All data produced here are available and can be produced upon request.

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
