# Peer review of "Exploring Patient Adherence to Post-Surgical Follow-Up in Pelvic Endometriosis"

_medicina, 2023, doi:10.3390/medicina59071210_

Round 1

Reviewer 1 Report

The manuscript is devoted to postoperative endometriosis, a significant medical and social problem. A sufficient number of observations over ten years were involved in the study. I liked the introduction section, where the research problem is described in detail. I believe this manuscript can be accepted for publication after minor revision.

I have the following minor comments:

1) it seems that Table 4 has missing data or should be restructured more understandably.

2) the Conclusions section needs to be improved by adding specific work results, as the text appears as part of a discussion.

Author Response

Cover letter 1

Yes, I explained  Table 4 better: : n is nr of patients, % is percentage of patients, S1,S3,S5,S7 the same questionnaire after 1, 3, 5 respectively 7 years of follow-up( they lost follow-up after 1 year).( Line 232-233)It is about Group A that lost the  follow-up after 1 year.We presented them in more detail because we wanted to understand the reason for the loss of follow-up.In table 5 we tried to compare the two groups, the one who lost follow-up with those who answered the questionnaire even after several years.

We added in the conclusions the results considered conclusive for the study.

Thank you.

Reviewer 2 Report

The great effort spent to conduct this research is greatly appreciated, however, kindly address the following comments: 

1. The title: since the main objective of the study was to address patients' follow up after surgery regarding different symptoms, i would suggest to change the title to show this, such as exploring long term endometriosis related symptoms after surgery 

2. It has been mentioned in the results that 453 patients completed first year follow up but 528 completed the second follow up, how can the second group who had completed 2 follow ups be greater than the first? 

3. in the paragraph of results: the table (173-178) is unclear? non-understandable

4. abbreviations in table should be mentioned below each table. 

for example: PMA (what is PMA)

5. Discussion should enriched further with papers that had previously investigated the outcome after endometriosis surgery 

6.  In line 323, it has been mentioned:  Our study found that surgery did not significantly improved overall pain levels especially group A. 

BUT is has been mentioned previously:

246-247: Moreover, the decrease  of intensity of pain reported after surgery between the two groups was significantly 247 higher for patients in group A (overall pain 63.8% from 95.6% to 31.8% vs. 56% from 93.5% to 37.5%, p<0.05; 

263-264: Post-surgical pain associated to endometriosis has a higher improvement for group A (with a drop from 82.9% to 55.1%) comparing to group B ( a drop from 67.9% to 52.6%),  with a decrease in the need of pain medication statistically significant.

It need to be explained further.

7. What is the pain level score mentioned in the following statement? 

298-299: Even if the decrease of intensity of pain reported after surgery between the two groups was significantly higher for patients in group A, as percentage of patients, pain level score improvement was lower for group A. 

Author Response

 Cover letter 2:

  • I changed the title as suggested.
  • Out of 2538 total number of surgeries, we managed to include 981 patients from which 453 (group A) patients replied just to the first questionnaire, 528 (group B) to the first and second, 445 to the first three questionnaires, and 356 carried out the entire follow-up schedule. So group A includes 445 patients, those who answered the questionnaire before surgery and one year after surgery and then lost the follow-up. Group B which includes 528 patients answered at least two questionnaires after surgery.( Line 169-172)
  • You are right and I changed it: Mean age of our patient was 32.52 with a SD Standard deviation of 6.87.(line 174)
  • The legend: n e nr of patients, % is percentage of patients, S1,S3,S5,S7 the same questionnaire after 1, 3, 5 respectively 7 years of follow-up, PMA( medically assisted procreation)
  • Discussion now contain more references about surgery than before and I also included a paragraph about follow-up after endometriosis surgery( line 406-411)
  • And 7) Line 246-247 more patients , in number and percentage, reported a decrease in pain intensity in group A, but the intensity was higher after surgery on the pain scale as I mentioned, with a new explanation about the scale: ; pain intensity with no treatment 2.11meaning 8.48 minus 6.37 vs. 1.9 meaning a drop from 7.89 to 5.99, but not significantly p>0,05; pain intensity with treatment 1.71 again 4.84 minus 3.13 vs. 1.28 mean intensity droped from 4.16 to 2.88,but not statisticaly significant p>0.05), on the pain scale 0 meaning no pain and 10 meaning extreme pain.( line251-255) Also:Pain during intercourse was reported significantly more often in group A compared to group B (prior to surgery 78.3% vs. 68.5%; 1 year post surgery 39.5% vs. 37.0%)( line 259-260).
  • Line 263-264 was from group B during years of follow-up and included by mistake.

Also you can see that general health, energy level, social functioning reported in group A was lower than in group B(line 281-296)

Line 298-299: Even if the decrease of intensity of pain reported after surgery between the two groups was significantly higher for patients in group A, as percentage and numbers of patients, pain level score improvement was lower for group A(mean intesity without treatment 6.37 in group A vs 5.99 in group B, mean intensity with treatment 3.13 versus 2.88) statistically significant, during menstrual cycles or chronic pain. Here I added the pain scale, 0 meaning no pain and 10 meaning extreme pain. I hope it is better now.

Line 323 is more a conclusion.(line 320-326).

We can speculate that the dissatisfaction of the patients in group A determined the abandon of follow-up, the lack of confidence in the treatment method. Thank you.
